# Survey of Frontline Police Officers’ Responses and Requirements in Psychiatric Emergency Situations

**DOI:** 10.3390/ijerph18010237

**Published:** 2020-12-30

**Authors:** Kyung Ja Lee, Kyunghee Lee, Yeong Mi Lee, Hyun Seok Choi

**Affiliations:** 1Department of Nursing, Milaelo Psychiatric Hospital, Gumi 39434, Korea; byoung7273@hanmail.net; 2College of Nursing, Keimyung University, Daegu 42601, Korea; 3Police Human Resources Development Institute, Asan 31540, Korea; lym5012@police.go.kr; 4Center for Educational Performance, Keimyung University, Daegu 42601, Korea; chsuk1@kmu.ac.kr

**Keywords:** frontline police officer, psychiatric, emergency

## Abstract

Police officers in South Korea can be summoned to incidents involving individuals with mental health problems. Therefore, for officers to communicate effectively in such situations, education is necessary. Accordingly, this study obtained frontline police officers’ perceptions of such educational programs and their suggestions regarding supplementary field manuals. Data were collected from 471 frontline police officers from 8 July until 9 August 2020. Data analysis incorporated frequency analysis, cross tabulation, text mining, and meaning network analysis. Participation in educational programs related to people with mental health problems depended on officers’ field experience with such persons (χ^2^ = 7.432, *p* = 0.006). Among officers who received educational programs, most expressed satisfaction with the programs (χ^2^ = 72.243, *p* < 0.001) and believed that these facilitated problem-solving (χ^2^ = 7.574, *p* = 0.023), improved understanding of people with mental health problems (χ^2^ = 10.220, *p* = 0.006), enabled better communication with such individuals (χ^2^ = 21.588, *p* < 0.001), and improved confidence in clarity of verbal expression in conversations with them (χ^2^ = 6.634, *p* = 0.036). An on-site response manual for communicating with people with mental health problems would represent an effective educational intervention to improve police judgment and responses.

## 1. Introduction

In South Korea, police officers must respond and intervene in sensitive, community safety-related issues that involve people with mental health problems. With an increasing number of individuals with mental health problems, and the complexity of relevant regulations and laws, cooperation between the police and mental health institutions has become very important to appropriately address the demands of people with mental illness and social demands [1]. Worldwide, there are increasingly many reports to the police of people with mental illness exhibiting severe symptoms; consequently, police response time and the use of physical force may be increasing in such situations [2].

In the United States in 2015, 124 of 462 persons killed by police gunfire were people with mental health problems. In most such cases, the police officers mobilized following a request for help from family, relatives, or friends of the mentally ill person [3]. In South Korea, according to a survey in 2018, crimes committed by people with mental health problems increased by 18.9%, from 1016 perpetrators in 2015 to 8343 in 2016 [4]. Despite an overall reduction in crimes over the previous five years, involvement of people with mental health problems in the criminal justice system is steadily increasing [5]. Police in many countries, and primarily the United States, have considered the best ways to interact with people with mental health problems. A typical approach is the Crisis Intervention Team (CIT) training model or the Memphis model, which was formulated through collaboration between academia and mental health workers [6]. These models aim to improve the safety of police officers and communities in psychiatric emergency situations, reduce crimes committed by people with mental health problems, and reduce stigma associated with mental illness. Active training is currently received by police officers in 45 states in the United States [7,8]. The models are designed to connect people with mental health problems to mental health institutions and improve the capability of such persons to leave the criminal justice system [6,9]. This has been successful in many law enforcement institutions worldwide [10,11]. It has been emphasized that one of the core factors for models to be successful is cooperation with mental health experts [1,12].

With an increasing number of police officers in South Korea encountering people with mental health problems, demands for measures to address crises involving people with mental health problems have also increased. Accordingly, a Psychiatric Emergency Response Manual that provides instructions for responders on how to manage psychiatric emergency situations was published jointly by the Ministry of Health and Welfare, the National Mental Health Center, the National Police Agency, and the Fire Department through policy [13]. This manual introduces the concept of mental illness, defines a psychiatric emergency, and presents intervention methods and on-site response methods. The manual was developed in cooperation with mental health experts and then distributed to relevant institutions. However, Choi and Lee detailed several problems with the manual [14]: first, the reliability and validity of the suggested situation assessment tool have not been adequately assessed. Second, fewer symptoms and types of mental illness are considered in the manual, and the classification of mental illness is very granular [14]. Police officers need to judge whether they should transfer a mentally ill person to hospital or arrest the person as a criminal breaching law. People with mental health problems in a state of emotional confusion may be at risk of self-harm or harming others. Accordingly, manuals that address police response in stressful situations take into consideration the officer making a rational judgment; adequate actions are not typically specifically prescribed in manuals [15]. Even when manuals by which to assess the need for emergency hospitalization have been published, frontline police officers continue to experience difficulties in psychiatric emergency situations. According to Ruiz and Miller, police officers have no time to carefully evaluate situations in most cases, and despite the stipulations of a manual, officers may adopt a forceful approach to quickly solve the situation; this approach can devolve into violence and injuries to both police officers and mentally ill persons [16]. Previous studies of South Korean psychiatric emergency situations indicate that police officers lack knowledge of the characteristic symptoms of people with mental illness and receive insufficient education regarding appropriate actions to take when interacting with such individuals. Consequently, frontline police officers consider responding to incidents involving people with mental health problems a burden, and they can be inactive in addressing such incidents [15,17].

In South Korea, according to “The third preliminary survey of the national mental health situation,” verification has not taken place of whether police officers can apply and use their knowledge of the classification standards when interacting with people with mental health problems [18]. Although the manual specifies that danger of self-harm or harming others is evaluated through cooperation with a specialized agent by contacting the area’s mental health promotion center, police officers experience difficulties using the evaluation tool, because there is a lack of personnel at each mental health center to support this procedure [18]. Accordingly, it is very difficult for mental health experts to provide timely support in psychiatric emergency situations. Therefore, police officers need to address situations involving people with mental health problems using their subjective judgment with few institutions/personnel from which/whom police officers can seek advice [15,17]. For police officers to effectively cope with psychiatric emergency situations, it is important that they have a basic understanding of mental illness, symptom judgment, and how to communicate with people with mental health problems. If systematic education in communication skills is not carried out, it will be difficult to appropriately interact with people with mental health problems, despite the existence of the aforementioned manuals and evaluation tools. Hence, the need for education programs based on the relevant practical work has been emphasized [14,15,17]. Nevertheless, relevant educational programs are scarce, as are relevant previous studies in South Korea. Educational programs that support police officers’ independent on-site response to people with mental health problems will not be successful without the active cooperation of mental health experts and psychiatric medical institutions [5,14]. Therefore, the need for adequate understanding and practical education with respect to mental illness extends beyond simply offering a manual or guidelines. Relevant information is limited to several previous studies, and so it is difficult for frontline police officers to properly address problems with on-site responses to psychiatric emergency situations.

The current study was conducted to provide baseline data regarding the response to psychiatric emergency situations of frontline police officers by considering police officers’ field experience with people with mental health problems in South Korea. Specifically, the study elucidated the priorities of officers and their uncertainties in psychiatric emergency situations, to improve their perceptions and behavioral intentions regarding people with mental health problems and thereby promote more appropriate on-site responses. The purpose of this study was as follows:Identify the level of participation in relevant educational programs related to people with mental health problems, depending on the police officers’ field experience with such persons.Assess police officers’ satisfaction with educational programs in terms of the programs supporting effective problem solving; facilitate understanding of, and responses to, people with mental health problems; provide knowledge on communication; and improve understanding of verbal expression characteristics of people with mental health problems.Identify the priority requirements of police officers in psychiatric emergency situations.

## 2. Materials and Methods

### 2.1. Study Design

This was a descriptive survey of frontline police officers’ perceptions of, and attitudes toward, people with mental health problems in psychiatric emergency situations, and officers’ requirements regarding educational support for such situations.

### 2.2. Participants

The target population of this study was South Korean frontline police officers, specifically patrol officers affiliated with 18 regional police agencies nationwide. To determine the required sample size, Raosoft Interform’s survey sample size calculator was used [19]. Given a ±4.5% maximum margin of error and a confidence level of 95%, the recommended sample size was 470 people. Data collection was conducted through collaboration with a researcher at the Police Human Resources Development Institute. The data of 471 police officers were finally analyzed.

### 2.3. Tools

The participants’ general characteristics were measured via nine questions that recorded gender, age, marital status, religion, work experience, on-site response satisfaction, experience with people with mental health problems, number of encounters with people with mental health problems, and experience of receiving educational programs related to mental illness. The police officers also detailed their social perceptions of people with mental health problems; respondents were classified according to whether they had participated in educational programs. The participants were also instructed to evaluate their satisfaction with the current on-site response manual for psychiatric emergency situations and to indicate their preferences for additions to the manual. Preferences were obtained using free responses. Questions on the on-site response manual asked whether field responses were performed according to the stipulations of the manual, and whether current educational programs are suitable for emergency situations. Whether educational programs need modification and what modifications are needed were assessed via 30 questions. To ensure the suitability, validity, and reliability of the questionnaire, it was revised and reviewed through advice from an expert group consisting of one professor of mental health nursing, two mental health experts, and one police administration professor. To test the implementation of the questionnaire, a preliminary survey was performed with 10 frontline police officers, after which the questions were revised as needed. Cronbach’s α was 0.83 in this study, which indicates acceptable internal consistency reliability.

### 2.4. Data Collection

Data collection was conducted after receiving approval from the Institutional Review Board of K University (IRB No: 40525-202004-HR-005-03). The principal and co-investigators of this study visited the Human Rights Sensitivity Education Center at the Police Human Resources Development Institute located in Asan, Chungnam, to explain the purpose of this study and to seek cooperation. After obtaining approval, the purpose and method of this study were explained in the introduction of the questionnaire. It was also specified that the participants could stop participating in this study at any time, the questionnaire responses would be handled anonymously, and the collected data would be used for only the study purpose. After written consent was obtained from participants, they could respond to the structured questionnaire directly using Google Forms Questionnaire. The questionnaire also provided contact information through which the participants could contact the researchers in the event of queries. The data collection period was from 8 July until 9 August 2020; the questionnaire survey targeted frontline police officers nationwide.

### 2.5. Data Analysis

The general characteristics of participants were described using frequencies and percentages. Social perceptions and attitudes toward people with mental health problems according to respondents’ participation in educational programs were assessed using crosstabulation analysis with standardized residuals. Preferences for changes to the on-site response manual were analyzed using text mining and semantic networks. Semantic network analysis is useful for understanding the flow of meaning contained in text through links among key words (nodes) in a network structure [20]. Through structural analysis of the role words play in relationship with concepts and whether words are arranged in a specific pattern, it is possible to understand the structure of specific meanings between concepts [21]. A semantic network is used when one is investigating knowledge that is best understood as a set of related concepts. In this study, the semantic network analysis refined and categorized the meaning of appropriate information using unstructured data [22]. Opinions of police officers were obtained regarding the difficulty of judging the symptoms of mentally ill persons and communicating with such persons. Keyword frequency of occurrence was visualized using a Word Cloud technique, and semantic network analysis identified meaningfully related words based on their simultaneous appearance in sentences.

## 3. Results

### 3.1. General Characteristics of Participants

There were 471 participants (89.8% male, 50 s or older 49.5%, married 71.5%). Of the 471 respondents who answered to the question “Satisfaction with on-site response,” less than half of the participants answered negatively (40.6%). Among the participants who answered to the question “Experience with people with mental health problems,” almost all of them answered “yes” (93.2%). But, over half of the respondents did not receive the educational programs (57.2%). The sociodemographic characteristics of the participants are summarized in Table 1.

### 3.2. Attitude during Field Responses According to Participation in Educational Programs

Most participants who had received educational programs pertaining to people with mental health problems were satisfied with the programs (χ^2^ = 72.243, *p* < 0.001). Further, most respondents who reported field encounters with people with mental health problems had participated in such educational programs (χ^2^ = 7.432, *p* = 0.006). Although many recommended the programs to colleagues, police officers who had received educational programs but had no field encounters more frequently did not recommend such programs (χ^2^ = 19.387, *p* < 0.001). Most police officers who had received educational programs believed that these helped them to effectively solve problems in interactions with people with mental health problems (χ^2^ = 7.574, *p* = 0.023). Among participants who had participated in educational programs, evaluation of moderate need for the programs was relatively frequent, while among participants who had not participated in such programs, evaluation of there being no need for the programs was relatively frequent (χ^2^ = 6.160, *p* = 0.046). Most participants who had received educational programs believed that such programs improved their understanding of, and responses to, people with mental health problems (χ^2^ = 10.220, *p* = 0.006); provided basic knowledge on communication with people with mental health problems (χ^2^ = 21.588, *p* < 0.001); and helped respondents to provide clear verbal expressions in conversations with people with mental health problems (χ^2^ = 6.634, *p* = 0.036; Table 2).

Table 3 reports the characteristics of participants who reported the perception of people with mental health problems. The average score was 5.38. Of the 467 respondents who answered to the question, the characteristics of the participants were summarized as follows: “Mental illness only affects the poor people” (88.4% No vs. 11.6% Yes), “people with mental illness are violent” (54.8% Yes vs. 45.2% No), “mental illness are caused by weak ego” (75.6% No vs. 24.4% Yes), “there is no perfect treatment for mental illness” (64.9% Yes vs. 35.1% No), “stigma and social prejudice associated with mental illness makes many people reluctant to contact people with mental health problems” (90.4% Yes vs. 9.6% No), “medication is useful for treating mental illness” (83.5% Yes vs. 16.5% No), “all individuals with mental health problems have the potential to become potential criminals” (54.6% Yes vs. 45.4% No), “all individuals with mental health problems mumble and talk to themselves” (74.7% No vs. 25.3% Yes).

### 3.3. Requirements for Interactions with People with Mental Health Problems and Suggested Additions to the On-Site Response Manual

#### 3.3.1. Suggested Requirements

When looking at high-frequency words, many opinions were presented related to police officers’ judgment and response, admission, and communication regarding programs. Semantic network analysis was used to identify the associated words based on the frequency with which words simultaneously appeared within the sentences. Links were revealed among police officers, intervention, and degree; people with mental health problems, response, and judgment; and hospital, referral, and admission (Figure 1).

In Figure 1, looking at the contents of the difficulty in judging and communicating the symptoms of the mentally ill, the words ‘police-intervention’ and ‘police-degree’ are used together in the sentence where police appear.

‘Mental illness-response’, ‘mental illness-judgment’, ‘mental illness-admission’, ‘admission-referral’, ‘mental illness-admission’, ‘admission-hospital’ were found to be used together in sentences containing mental illness.

#### 3.3.2. Suggested Additions to the On-Site Response Manual

Words with high frequency referred to the hospitalization system, clarity, cooperation, and referral to other institutions. Semantic network analysis of interview content identified simultaneous occurrences within sentences of medical institution and admission procedure; cooperation, related institutions, and family; hospitalization system and human rights; and on-site response and referral (Figure 2). Police officers appeared to want additions to the on-site response manual that addressed psychiatric emergencies. The connections among words (i.e., {medical institutions, hospitalizations, procedures}, {hospitalization system, human rights}, {on-site response, referral}) suggested the need for establishing a hospitalization system and simplifying procedures for cooperation with medical institutions, between related institutions and families, and between the on-site response system and referrals.

In Figure 2, if you look at the opinions on the matters that you would like to further supplement the “Psychiatric Emergency Site Response Manual” currently in use,

The words such as ‘clear,’ ‘hospitalization’, ‘cooperation’, other institutions, referral, medical institution, police officer, manual, response, task, judgment, afterwards, and compulsion were frequently used.

Looking at the items that you want to further supplement the on-site response manual for psychiatric emergencies, the words ‘cooperation-other institutions’ and ‘cooperation-family, are used together in the sentence where cooperation appears.

In the sentence where ‘response system’ appears, the words ‘response system-manual’ and ‘response system-referral’ are used together,

It was found that the words procedure-medical institution, medical institution-hospitalization, hospitalization-human rights are used together.

## 4. Discussion

More than 90% of respondents reported field experience with people with mental health problems, and 60% of officers reported five or more experiences. First, most participants with such field experience had participated in educational programs related to people with mental health problems. Among participants who had participated in these educational programs, most were satisfied with the programs. The results confirmed that the police officers were highly interested in education regarding emergency situations and that they perceived that the methods used when interacting with people with mental health problems could be improved, as shown in Table 3. According to the CIT model, cooperation with, and interventions by, mental health institutions are needed. In this model, police officers receive 40 h of training provided by mental health doctors, other mental health experts, and police trainers. The training is instructive, experiential, and includes practical skills/scenario-based training [1,6]. A previous study indicated that the CIT program reduced inappropriate law enforcement against mentally ill persons and was effective in improving the safety of mental health services. A subsequent study found that the crime rate among people with mental health problems was low [6].

Second, police officers with experience of educational programs considered such programs more necessary to inform field actions than those who had not received such education. A prior study reported that police officers with education regarding people with mental health problems perceived they could more safely respond to psychiatric crisis situations than those without such education. Further, the education had positive effects on police officers’ confidence in their capability to respond to people with mental health problems, communication technique, and the overall relationship between attitude and department [23].

Third, among police officers who had received educational programs, a relatively large proportion believed that they understood and appropriately responded to people with mental health problems, held basic knowledge on how to communicate appropriately with such individuals, and could utilize clear verbal expressions in conversation with them. If police officers experience educational programs even once, subsequent access to education appears to become easier. Police officers with experience of interacting with people with mental health problems had relatively positive perceptions of their field responses to such persons, and their empathy and understanding of people with mental health problems improved; thus, social perceptions or attitudes toward people with mental health problems were positively changed.

In South Korea, the 2020 Police Human Resources Development Institute organized courses for police officers in the field to promote understanding and responding to mentally ill patients. These two courses are a combination of online and classroom education, with 35 h per week in groups of 15 people. The main contents are based on the on-site response manual for psychiatric emergencies. The manual contains details on the nature of the mentally ill patients at high-risk who may need intervention from on-site police officers, an overview of emergency interventions, and step-by-step countermeasures. The main psychiatric concepts are that police officers intervene in cases of mental illness, specifically emergency crises that include suicide-related incidents and attempts, and those that involve risk of self-harm or harm, such persons with alcohol use disorders.

Additionally, police officers receive continuing training in various forms, such as oral or documentary delivery of problems and practical guidelines related to the response, when cases occur that highlight social issues related to people with mental health problems or cases that can be used as a reference [24].

In the case of on-site response, intensive and professional education via medical experts is required, because of the difficulty of communicating with people with mental health problems. As shown in Figure 1, police officers considered determining the presence of mental illness to be one of the demanding aspects of on-site response. As shown in Table 2, 269 officers (57.2%) did not believe that they had sufficient basic knowledge to effectively communicate with people with mental health problems. It is perhaps unreasonable to expect police to provide therapeutic interventions. It is difficult for police to judge risk factors in an emergent crisis situation, or to determine the likelihood of future threats through communication with mentally ill persons, especially if risk factors have been resolved before they arrive at the scene.

In terms of assessing medical symptoms, prior education on the symptoms of mental illness would be necessary through illustration of cases by mental health experts, such a psychiatrists and psychiatric nurses. Such education could be provided in person or online and should be a compulsory component of police education. Analysis of the semantic network revealed linkage between the degree of intervention and the responsibilities of the police officers, and among assessment of people with mental health problems, the response method, and referral and admission to hospital (Figure 1).

The suggested requirements for the field response manual for psychiatric emergencies showed connections between establishing a hospitalization system and simplification of procedures for cooperation with medical institutions; and connection and cooperation among related institutions and families, the on-site response system, and referral system (Figure 2). It is necessary to develop scenario-based, simulation education and training by experts in various fields related to police response to situations involving mentally ill persons.

As in the United States, the police in South Korea should develop a CIT or Memphis training model centered on academics and practitioners to maintain continuous training and education for field workers. Fortunately, the National Police Human Resources Development Institute is beginning to recognize the importance of responding appropriately to people with mental health problems. For example, a course was established in 2020 to address understanding and responding to people with mental health problems, including the appropriate use of physical force. This course involves simulation of scenarios involving people with mental health problems. The course was developed not only with the assistance of local police experts with extensive experience in the field, but also with members of external organizations, such as 119, mental health centers, and psychiatric facilities. To enhance the ability of police officers to respond to various cases, curriculum design should be discussed and modified based on detailed feedback. It has been emphasized that one of the key elements of the CIT model is cooperation with community partners including mental health professionals [1,12].

The priority requirements suggested by the police officers were as follows. First, the most important requirement in psychiatric emergency situations was a standard by which to assess people with mental health problems. The need for professional knowledge was emphasized to determine whether the suspect is mentally ill and to formulate a suitable response and follow-up.

The police officers emphasized that they needed to assess whether they should commit an individual with mental illness to a hospital by regarding them as a threat to their own or others’ life or property, versus arresting them as a criminal violating laws [15]. In psychiatric emergencies, frontline police officers are responsible for transferring persons at risk of self-harm to a psychiatric institution. However, police tend to be reluctant to intervene in these situations because of the risk of becoming involved in human rights issues. Community mental health centers lack mental health professional human resources, budget, and authority in South Korea. Therefore, cooperation with community mental health teams in emergency situations involving severely mentally impaired persons is not well established [25]. In this context, the absence of systematic education by which to make objective judgments in psychiatric emergency situations was considered an important deficit.

Second, the need was emphasized for a method to cope with psychiatric emergency situations, including a need for hospitalization, and cooperation and linkage with hospitals. Although there exists a relevant manual, the police officers considered it difficult to use the manual in practical work, and that it was difficult to collaborate with mental health professionals during each psychiatric emergency. Suggested additions to the manual for addressing psychiatric emergency situations included detailing cooperation with other institutions.

Third, the need for communication methods applicable to psychiatric emergency situations was emphasized. Systematic education for police officers in communication skills is required, which centers on practical skills needed to interact effectively with people with mental health problems in crisis situations.

In summary, this study provided the following information. First, objective evidence was provided of the need for practical education regarding the symptoms of mental illness to encourage police officers to view people with mental health problems from a nursing perspective. Second, this study indicated that perceptions of, and attitudes toward, people with mental health problems changed contingent on receiving educational programs that adopted a nursing perspective of mental illness. Third, this study elucidated police officers’ field application of the knowledge gained from educational programs that adopted a nursing perspective.

This study has value in that it characterized the on-site responses of frontline police officers in psychiatric emergency situations using free responses and multiple-choice questions. The data confirmed frontline police officers’ educational program experience and requirements, and presented suggested content to be included in education for police officers regarding psychiatric emergency situations. Note that in this study, all word cloud and semantic network analyses were conducted in South Korean; however, they were translated into English for this article. Therefore, vocabulary differences may present limitations in conveying the correct meaning.

## 5. Conclusions

Frontline police officers may encounter people with mental health problems in law enforcement situations and they should therefore have the capability to cope with such situations. This study considered education as a method to improve police coping in psychiatric emergency situations by identifying educational program experience and the educational requirements of frontline police officers. Specifically, judgment and communication skills in interactions with people with mental health problems were highlighted as practical requirements. This study suggests the need for educational programs that improve attitudes toward people with mental health problems by reducing negative perceptions of this group and enhancing field responses to such persons. This study also suggests the need for research into practical educational programs that address how to assess people with mental health problems and communicate with them; these represent the most pressing educational needs.

## Figures and Tables

**Figure 1 ijerph-18-00237-f001:**
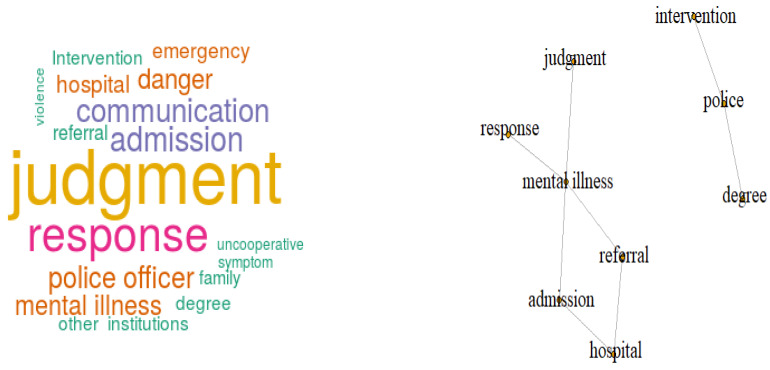
Word cloud analysis regarding the most preferred requirements.

**Figure 2 ijerph-18-00237-f002:**
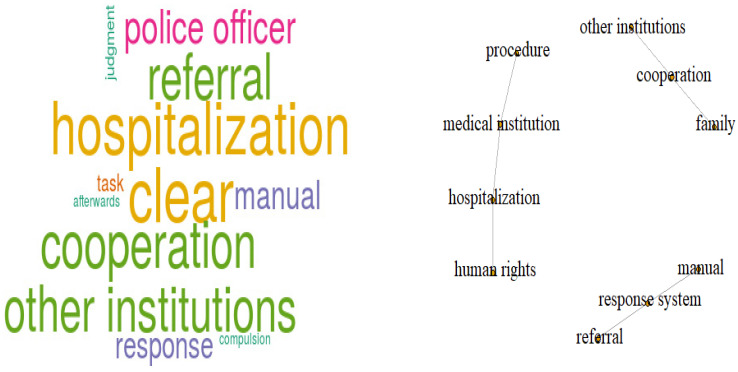
Word cloud analysis of suggestions for additions to the on-site response manual.

**Table 1 ijerph-18-00237-t001:** General characteristics of the participants (N = 471).

Variable	Category	Frequency	Percentage
Gender	M	423	89.8
	F	48	10.2
Age	20 s	62	13.2
	30 s	105	22.3
	40 s	71	15.1
	50 s or older	233	49.5
Marital status	Never married	128	27.2
	Married	337	71.5
	Divorced	6	1.3
Religion	Christianity	76	16.1
	Buddhism	120	25.5
	Catholicism	40	8.5
	Other	235	49.9
Years of work experience	<3	79	16.8
	3 ≤ 5	31	6.6
	5 ≤ 10	53	11.3
	10 ≤ 15	36	7.6
	>15	272	57.7
Satisfaction with on-site response	Yes	37	7.9
	To some degree	243	51.6
	No	191	40.6
Experience with people with mental health problems	Yes	439	93.2
	No	32	6.8
No. of experiences involving people with mental health problems	1 ≤ 3	86	19.2
	3 ≤ 5	74	16.5
	>5	288	64.3
Received educational programs	Yes	201	42.8
	No	269	57.2

**Table 2 ijerph-18-00237-t002:** Attitudes of police officers toward people with mental health problems depending on participation of the former in educational programs (frequencies and standardized residuals).

Items	PIEP	Yes	Moderate	No	Total	χ^2^	*p*
PIEP and satisfaction with educational programs related to people with mental health problems	Yes	17 (1.96)	134 (3.28)	50 (−4.16)	201	72.243	<0.001
	No	2 (−2.21)	47 (−3.68)	110 (4.68)	159		
PIEP and recommendation of programs	Yes	35 (−1.11)	114 (2.14)	51 (−1.67)	200	19.387	<0.001
	No	41 (1.24)	54 (−2.39)	65 (1.87)	160		
PIEP and its help in effective problem solving	Yes	24 (0.1)	125 (1.06)	52 (−1.48)	201	7.574	=0.023
	No	18 (−0.11)	78 (−1.2)	62 (1.67)	158		
PIEP and need for the programs	Yes	133 (−0.68)	60 (1.54)	8 (−0.82)	201	6.160	=0.046
	No	197 (0.59)	55 (−1.33)	17 (0.71)	269		
PIEP and understanding of expressions of people with mental health problems	Yes	29 (1.23)	109 (1.03)	63 (−1.81)	201	10.220	=0.006
	No	25 (−1.06)	122 (−0.89)	122 (1.57)	269		
PIEP and basic communication knowledge	Yes	43 (2.88)	97 (0.26)	61 (−1.99)	201	21.588	<0.001
	No	22 (−2.49)	124 (−0.22)	123 (1.72)	269		
PIEP and clear verbal expression	Yes	36 (1.37)	114 (0.31)	51 (−1.35)	201	6.634	= 0.036
	No	31 (−1.19)	145 (−0.27)	93 (1.17)	269		
PIEP and worries about understanding	Yes	79 (0.79)	98 (−0.5)	24 (−0.33)	201	1.716	=0.424
	No	90 (−0.68)	143 (0.43)	36 (0.28)	269		
PIEP and belief in skills to cope with various problems	Yes	56 (−0.92)	94 (−0.4)	51 (1.78)	201	7.282	=0.026
	No	92 (0.79)	135 (0.34)	42 (−1.54)	269		
PIEP and capability to cope with crises	Yes	44 (2.2)	128 (0.32)	29 (−2.37)	201	18.444	<0.001
	No	30 (−1.9)	163 (−0.28)	76 (2.05)	269		
PIEP and difficulties coping with people with mental health problems	Yes	108 (0.1)	79 (0.18)	14 (−0.66)	201	0.828	=0.661
	No	142 (−0.09)	102 (−0.16)	25 (0.57)	269		
PIEP and belief that mental illness affects only poor people	Yes	20 (−0.73)		181 (0.26)	201	1.043	=0.307
	No	35 (0.63)		234 (−0.23)	269		
PIEP and belief that people with mental health problems are violent	Yes	106 (−0.41)		95 (0.46)	201	0.660	=0.417
	No	152 (0.36)		117 (−0.39)	269		
PIEP and belief that mental illness occurs because of weakness	Yes	49 (−0.03)		152 (0.01)	201	0.002	=0.969
	No	66 (0.02)		203 (−0.01)	269		
PIEP and belief there is no perfect treatment for mental illness	Yes	138 (0.7)		63 (−0.95)	201	2.430	=0.119
	No	166 (−0.61)		103 (0.82)	269		
PIEP and the belief that many people are reluctant to contact people with mental health problems	Yes	182 (0.05)		19 (−0.15)	201	0.045	=0.833
	No	242 (−0.04)		27 (0.13)	269		
PIEP and the belief that drug treatment is useful for mental illness treatment	Yes	163 (−0.36)		38 (0.8)	201	1.354	=0.245
	No	229 (0.31)		40 (−0.69)	269		
PIEP and belief that people with mental health problems are potential criminals	Yes	112 (0.2)		89 (−0.22)	201	0.153	=0.695
	No	145 (−0.17)		124 (0.19)	269		
PIEP and the belief that the mentally ill mutter and mumble to themselves	Yes	52 (0.16)		149 (−0.09)	201	0.056	=0.812
	No	67 (−0.13)		202 (0.08)	269		
PIEP depending on field experience with people with mental health problems	Yes	195 (0.53		6 (−1.99)	439	7.432	=0.006
	No	244 (−0.46)		25 (1.72)	31		

PIEP: Participation in educational programs.

**Table 3 ijerph-18-00237-t003:** Social perceptions of people with mental health problems by police officers (N = 467).

Thoughts on People with Mental Illness	Yes		No		M	SD
	n	%	n	%		
1. Mental illness only affects the poor people.	54	11.6	413	88.4	0.88	0.32
2. People with mental illness are violent.	256	54.8	211	45.2	0.45	0.50
3. Mental illnesses are caused by weak ego.	114	24.4	353	75.6	0.76	0.43
4. There is no perfect treatment for mental illness.	303	64.9	164	35.1	0.35	0.48
5. Stigma and social prejudice associated with mental illness makes many people reluctant to contact people with mental health problems.	422	90.4	45	9.6	0.90	0.30
6. Medication is useful for treating mental illness.	390	83.5	77	16.5	0.84	0.37
7. All individuals with mental health problems have the potential to become potential criminals.	255	54.6	212	45.4	0.45	0.50
8. All individuals with mental health problems mumble and talk to themselves.	118	25.3	349	74.7	0.75	0.44
Total					5.38	1.44

## Data Availability

We do write a research with caution without falling into the trap of plagiarism.

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
