# Peer review of "Survey of Frontline Police Officers’ Responses and Requirements in Psychiatric Emergency Situations"

_ijerph, 2020, doi:10.3390/ijerph18010237_

Round 1

Reviewer 1 Report

Dear Authors,

I appreciate your effort to improve the manuscript and to reply my comments.

Author Response

No additional comments were provided in this round. We thank you for your careful consideration of our manuscript throughout the review process.

Your point of needing to improve our English more qualitatively made me somewhat embarrassed. The sentences have been refined continuously by professional native speakers. If you would point out the errors in the sentence, we would like to do our best to supplement them

Reviewer 2 Report

Thank you for you considered responses to the original review. This article is much improved. 

Thank you for the additions to the manuscript. It is much improved. 

Two minor issues are outstanding:

(1) it would be helpful if the authors noted whether or not ethics approval was required for the study. 

(2) the introduction (line 34) notes increased reports of people exhibiting severe symtoms worldwide- then states " consequently, police response time and the use of physical force are increasing in such situations". There is not an established causal relationship between these two phenomenon- certainly not on a world scale. Please consider modifying this sentence.   

Author Response

We thank you for your careful consideration of our manuscript throughout the review process.

(1) It would be helpful if the authors noted whether or not ethics approval was required for the study.

Details of ethical approval are provided under 2.4 Data Collection.

(2) the introduction (line 34) notes increased reports of people exhibiting severe symptoms worldwide- then states " consequently, police response time and the use of physical force are increasing in such situations". There is not an established causal relationship between these two phenomenon- certainly not on a world scale. Please consider modifying this sentence.

Your point is appreciated. The implication of causality has been removed from the sentence, as follows:

Worldwide, there are increasingly many reports to the police of people with mental illness exhibiting severe symptoms. Police response time and the use of physical force may be increasing in such situations.

This manuscript is a resubmission of an earlier submission. The following is a list of the peer review reports and author responses from that submission.

Round 1

Reviewer 1 Report

Dear Authors,

I really appreciate your manuscript, in most part of the sections it was well-written.

My principal concern was about the Semantic network analysis. The method was not described in data analysis section, I suggest you to read the following paper:

Drieger, P. (2013). Semantic network analysis as a method for visual text analytics. Procedia-social and behavioral sciences79(2013), 4-17.

In Results section, data from Semantic network analysis must be described as suggested by Drieger (2013 - see above). You have 471 interviews, I think that data from them are an important source of information about the police officers’ responses to psychiatric emergency situations.

Since you could use other type of statistical analysis (for example, content analysis with the support of statistical programs, such as T-Lab, Alceste and so on), it's very important to explain the method used to analyse interviews and argue this choice.

I suggest to rewrite Results and consequently Discussion sections. 

Moreover:

  • table 1: there was a mistake, the number 30s was ripeated
  • table 2: please, provide the translation of the first word in the table.

Author Response

Thank you for your feedback very much!!

1. The method was not described in data analysis section, I suggest you to read the following paper:

Drieger, P. (2013). Semantic network analysis as a method for visual text analytics. Procedia-social and behavioral sciences79(2013), 4-17.

In Results section, data from Semantic network analysis must be described as suggested by Drieger (2013 - see above). You have 471 interviews, I think that data from them are an important source of information about the police officers’ responses to psychiatric emergency situations.

Please provide your response for point 1. (line 316 and line 474).

2.

  • table 1: there was a mistake, the number 30s was ripeated
  • table 2: please, provide the translation of the first word in the table. 

Please provide your response for point 2. (line 197 and line 217-220).

Reviewer 2 Report

Where there are exact quotes the exact page number should be shown.

The reasons why the variables were chosen should be explained in detail. (e.g. why was a marriage or religion an important variable?)

With training on mental health issues, it may be of value to identify the length of the course undertaken and whether there are regular refresher courses given.

It may be of value if the types of mental illnesses, and their frequency, dealt with by officers are identified. This may direct the content of mental health training provided to officers.

Author Response

Thank you for your feedback very much!!

  1. Where there are exact quotes the exact page number should be shown.

    Please provide for point 1 (  line 423-477)                      

2. The reasons why the variables were chosen should be explained in detail. (e.g. why was a marriage or religion an important variable?)

Because we think the chacteristics of this sample should be explained minimally.

3. With training on mental health issues, it may be of value to identify the length of the course undertaken and whether there are regular refresher courses given.

Please provide for point 3 (line 272-276)

3. It may be of value if the types of mental illnesses, and their frequency, dealt with by officers are identified. This may direct the content of mental health training provided to officers.

Thanks!! I agree with your point. I will make sure to reflect it in future research.

Reviewer 3 Report

This research has the potential to be the basis of a very intersting article. It is difficult to judge the true merit because there is insufficient detail about the context of the research and the methods used. It seems that the project is about police satisfaction with a particular training program but this is not entirely clear. If that is the case, it is not clear how the research connects with the global issue of improving police interactions with people with mental illness.  More needs to be done to bring out the significance of this work. 

OVERALL COMMENT

The authors have provided a paper on a topic which is of considerable interest throughout the world. How police officers respond to incidents involving those who may have or appear to have mental health problems is a global issue of some import.

The circumstance in which such interactions occur is likely to differ. How police forces are configured, what role they play and how they are empowered is relevant. It would have been helpful if the authors had identified the specific jurisdiction-Korea- at the outset and provided some background information about the police force and the specific legal, policy and practice framework that governs the interaction of police with individuals that may involve mental health issues.  

The strength of the paper is its examination of police attitudes to specific mental health training. However, there is no description of the training package itself. Detailed analysis of the kind of training being assessed would have been helpful. Some of the comments in the paper seem to suggest that the framework developed in Korea envisages interaction or co-operation between police and mental health professionals. It was not clear whether this was a strategy ‘on the ground’ that is embedded in the Manual or merely that training was provided by mental health professionals.  Too little is explained. If the Manual is a unique approach to police training that is being developed in Korea it warrants a richer description. This would be of greater interest to international readership.

DETAILED COMMENT

Abstract: please indicate the jurisdictional country in which the research was undertaken in abstract.

Introduction

The paper could be more accurately framed. Please consider changing the word ‘must’ in the opening line. It is more accurate to say there is an increased likelihood that police will be called to incidents that involve individuals who are or appear to be affected by mental health problems. The next line states that there is or should be cooperation between police and mental health institutions. This may reflect an operative assumption in Korea but does not accurate accurately reflect the relationship between police and mental health services elsewhere. Following that sentence, there is an indication that the police interactions are those where police are all called to incidents. Elsewhere, there are many ways in which police become involved in incidents.  The paper is not clear whether it is referring only to those incidents where mental illness is already identified by the reporting party or encompasses other interactions. How the incidents to which the paper refers are defined as important. These and other generalisations limit the merit of the subsequent analysis.

Paragraph 2 provides a reasonable summary of literature about the need for police training. It would have been helpful to provide a more detailed account of the Memphis model, and what role understanding of that model has played in developing the approach used in Korea.

Paragraph 3 explains the development of the crisis response manual in Korea. A description of the Korean approach, including its genesis aims and objectives, how it compares with the Memphis model, and what models are used internationally would have been helpful here.

It is difficult to make sense of worth of the findings without a clear understanding of the applicable law in Korea, the formation and role of the police force, and their powers with respect to such incidents. It seems likely that practices for policing education are of enough diversity to warrant a more detailed explanation for an international audience.

This section of the paper also contains some unjustified and un-referenced assertions. For example: commencing at line 87:

“For police officers to effectively cope with psychiatric emergency situations, it is important that they have a basic understanding of mental illness, symptom judgment, and how to communicate with mentally ill persons.” 

It may be a guiding assumption in the Korean model, but it is not clear from the paper whether this is the approach used in Memphis or in other jurisdictions. This is not explained by the authors. In Australia for example, where there are similar problems with the police responses to those with mental health problems, the emphasis for best practice is on how to communicate with individuals in distress in ways that de-escalate the potential vines and harm. That is NOT an approach which is reliant on police officers developing an understanding of mental illness or symptom judgement. A more thorough analysis of the international literature on policing and mental health education may have improved the paper.

In summary, the introduction frames the research in a way that overstates its significance. The parameters of the study, and the research questions it poses, are in fact very narrow. The paper seeks to quantify the number of participants in training, identify whether participants were satisfied with the training, identify whether participants knowledge and expertise were improved, and identify what police thought was important.  There is, however, nothing in the methodology or study design that allows conclusion to be drawn about whether expertise has been improved.  

Material and methods

The summary of the study design at 2.1 is a more accurate description of the study outlined in the introduction. However, the discussion about the ‘tools’ suggests that the study is measuring how well the education assists police officers to adhere to a practice Manual.  A deeper explanation of what the study is doing, how it relates to the manual and how the tools relate to the research questions would be helpful.

The description of the cohort at 3.1 is sound.

The analysis at 3.2 is misleading. It is entitled social perceptions and attitudes towards people with mental illness. A brief look at the discussion and the material presented in table 2 shows that this analysis is about police officers’ attitudes to training and not about social attitudes to people with mental illness.

There are some interesting components presented in the findings, but they are not well presented. For example, as is made clear in the discussion there are findings about police officers’ thoughts about the limitations of the training. This data should be present in this- the findings section.

Discussion

The discussion section indicates that while there was generally a positive response many police officers thought that the training could be improved. The structure of the discussion is unusual. The usual approach is to present the literature first-what has been found already- and then present the results. And discuss. Once again, the presentation of the paper detracts from its potential impact.

Finally, there is no information about whether ethics approval or permissions were necessary and obtained by the research team.

Author Response

OVERALL COMMENT

1.The authors have provided a paper on a topic which is of considerable interest throughout the world. How police officers respond to incidents involving those who may have or appear to have mental health problems is a global issue of some import.

Thank you very much!

2.The circumstance in which such interactions occur is likely to differ. How police forces are configured, what role they play and how they are empowered is relevant. It would have been helpful if the authors had identified the specific jurisdiction-Korea- at the outset and provided some background information about the police force and the specific legal, policy and practice framework that governs the interaction of police with individuals that may involve mental health issues.  

Please provide for point 2. (line 297-313)

3.The strength of the paper is its examination of police attitudes to specific mental health training. However, there is no description of the training package itself. Detailed analysis of the kind of training being assessed would have been helpful. Some of the comments in the paper seem to suggest that the framework developed in Korea envisages interaction or co-operation between police and mental health professionals. It was not clear whether this was a strategy ‘on the ground’ that is embedded in the Manual or merely that training was provided by mental health professionals.  Too little is explained. If the Manual is a unique approach to police training that is being developed in Korea it warrants a richer description. This would be of greater interest to international readership.

Please provide for point 3. (line 297-313)

DETAILED COMMENT

1.Abstract: please indicate the jurisdictional country in which the research was undertaken in abstract.

Please provide for point 1. (line 11)

2.

The paper could be more accurately framed. Please consider changing the word ‘must’ in the opening line. It is more accurate to say there is an increased likelihood that police will be called to incidents that involve individuals who are or appear to be affected by mental health problems. The next line states that there is or should be cooperation between police and mental health institutions. This may reflect an operative assumption in Korea but does not accurate accurately reflect the relationship between police and mental health services elsewhere. Following that sentence, there is an indication that the police interactions are those where police are all called to incidents. Elsewhere, there are many ways in which police become involved in incidents.  The paper is not clear whether it is referring only to those incidents where mental illness is already identified by the reporting party or encompasses other interactions. How the incidents to which the paper refers are defined as important. These and other generalisations limit the merit of the subsequent analysis.

Please provide for point2, I have revised the text based on your comments. Thanks.

3. Paragraph 2 provides a reasonable summary of literature about the need for police training. It would have been helpful to provide a more detailed account of the Memphis model, and what role understanding of that model has played in developing the approach used in Korea.

Please provide for point3.(line 47-77)

4. Paragraph 3 explains the development of the crisis response manual in Korea. A description of the Korean approach, including its genesis aims and objectives, how it compares with the Memphis model, and what models are used internationally would have been helpful here.

Please provide for point4.(line 47-77)

5. It is difficult to make sense of worth of the findings without a clear understanding of the applicable law in Korea, the formation and role of the police force, and their powers with respect to such incidents. It seems likely that practices for policing education are of enough diversity to warrant a more detailed explanation for an international audience.

This section of the paper also contains some unjustified and un-referenced assertions. For example: commencing at line 87:

“For police officers to effectively cope with psychiatric emergency situations, it is important that they have a basic understanding of mental illness, symptom judgment, and how to communicate with mentally ill persons.” 

It may be a guiding assumption in the Korean model, but it is not clear from the paper whether this is the approach used in Memphis or in other jurisdictions. This is not explained by the authors. In Australia for example, where there are similar problems with the police responses to those with mental health problems, the emphasis for best practice is on how to communicate with individuals in distress in ways that de-escalate the potential vines and harm. That is NOT an approach which is reliant on police officers developing an understanding of mental illness or symptom judgement. A more thorough analysis of the international literature on policing and mental health education may have improved the paper.

Please provide for point 5, In korea, (First of all, compared to the total number of local police officers, the number of people exposed to education is very small. And, there is no independent course for the mentally ill, and it is organized into one subject.In the case of on-site response, intensive and professional education through medical experts is necessary in relation to the difficulty of communicating with people with mental illness). 

6. In summary, the introduction frames the research in a way that overstates its significance. The parameters of the study, and the research questions it poses, are in fact very narrow. The paper seeks to quantify the number of participants in training, identify whether participants were satisfied with the training, identify whether participants knowledge and expertise were improved, and identify what police thought was important.  There is, however, nothing in the methodology or study design that allows conclusion to be drawn about whether expertise has been improved.  

Please provide for point 6.(Please could review methods and discussion again?)

Material and methods

The summary of the study design at 2.1 is a more accurate description of the study outlined in the introduction. However, the discussion about the ‘tools’ suggests that the study is measuring how well the education assists police officers to adhere to a practice Manual.  A deeper explanation of what the study is doing, how it relates to the manual and how the tools relate to the research questions would be helpful.

The description of the cohort at 3.1 is sound.

1.The analysis at 3.2 is misleading. It is entitled social perceptions and attitudes towards people with mental illness. A brief look at the discussion and the material presented in table 2 shows that this analysis is about police officers’ attitudes to training and not about social attitudes to people with mental illness.

Please provide for point 1. (line 217)

2. There are some interesting components presented in the findings, but they are not well presented. For example, as is made clear in the discussion there are findings about police officers’ thoughts about the limitations of the training. This data should be present in this- the findings section.

Please provide for point 2. (Please notify the part of resuls and discussions)

Discussion

1.The discussion section indicates that while there was generally a positive response many police officers thought that the training could be improved. The structure of the discussion is unusual. The usual approach is to present the literature first-what has been found already- and then present the results. And discuss. Once again, the presentation of the paper detracts from its potential impact.

Please provide for point 1. (I have revised the part of discussion based on your comments. Thanks.)

2. Finally, there is no information about whether ethics approval or permissions were necessary and obtained by the research team.

Please provide for point 2. (line 160-161)

Reviewer 4 Report

Briefly, this manuscript presented a survey of frontline police officers’ perception of individuals with mental health problems in emergency situations and educational needs for supporting the officers when encountering such situations. The topic is significant, and there is a clear need to understand and address the influence of perceptions and attitudes of police force towards individuals with mental health problem experiencing a crisis as they are usually the first point of contact.

Some review points:

  1. The graphical representation of figure 2 is difficult to comprehend the findings.
  2. A broader issue is that there is a lack of discussion of study weaknesses
  3. Another pressing concern is the lack of objectivity in regard to concept of ‘psychiatric emergency”. Did the authors collect data on the level of interaction between the police officers or individuals experiencing a psychiatric emergency? Was it a psychotic episode, attempting suicide, panic attacks? This brings in the issue of the definition of ‘psychiatric emergency’. Each psychiatric emergency requires a different approach – has this been acknowledged or thought through? Are the perceptions of police officers of individuals with mental health problems different across diagnoses? For example, would a police officer respond the same towards an individual experiencing a psychotic episode compared to an individual attempting to harm themselves (suicide, substance abuse etc.?
  4. Stigma is a huge issue when it comes to police and mental health. I would recommend looking into this as this was not given much attention. It could give a perspective on how educational programs can be modified to support the police force in managing/approaching those undergoing psychiatric emergencies.

There is room for further analysis, which is needed to make this manuscript a more original contribution to the current state of the field.

Author Response

Briefly, this manuscript presented a survey of frontline police officers’ perception of individuals with mental health problems in emergency situations and educational needs for supporting the officers when encountering such situations. The topic is significant, and there is a clear need to understand and address the influence of perceptions and attitudes of police force towards individuals with mental health problem experiencing a crisis as they are usually the first point of contact.

Some review points:

1.The graphical representation of figure 2 is difficult to comprehend the findings.

Please provide for point 1.(Thank you for your valuable comments. I have revised from line 240 to line 263 based on your comments.)

2. A broader issue is that there is a lack of discussion of study weaknesses

Please provide for point 2.( Ihave revised from line 264 to line 400 based on your comment.)

3. Another pressing concern is the lack of objectivity in regard to concept of ‘psychiatric emergency”. Did the authors collect data on the level of interaction between the police officers or individuals experiencing a psychiatric emergency? Was it a psychotic episode, attempting suicide, panic attacks? This brings in the issue of the definition of ‘psychiatric emergency’. Each psychiatric emergency requires a different approach – has this been acknowledged or thought through? Are the perceptions of police officers of individuals with mental health problems different across diagnoses? For example, would a police officer respond the same towards an individual experiencing a psychotic episode compared to an individual attempting to harm themselves (suicide, substance abuse etc.?

Please provide for point 3.(From the perspective of the police, a psychiatric emergency is considered a safety issue rather than judging based on the dignosis. A person who threatens the life or property of oneself or others in a state of confusion should be considered as a mentally ill in need of treatment, and should be sent to the hospital, or as a criminal who violates the law, and should be judged as to whether to arrest. Moreover, front-line police officers are forced to face a situation where they must make quick and difficult judgments as to how to intervene in an urgent situation. However, since the field of work of frontline police officers is very broad and handles most cases, there is a limit to the professional response to psychiatric emergency situations.)

4.Stigma is a huge issue when it comes to police and mental health. I would recommend looking into this as this was not given much attention. It could give a perspective on how educational programs can be modified to support the police force in managing/approaching those undergoing psychiatric emergencies.

Please provide for point 4.(line 388-30)

5. There is room for further analysis, which is needed to make this manuscript a more original contribution to the current state of the field

Please provide for point 5.(line 221-263)

Round 2

Reviewer 1 Report

Dear Authors,

I appreciate your effort in improving your manuscript. However, I have some concerns about the method: it was not well explained. Why did you choose this method and not other? I suggest to insert a description of the Semantic network analysis (line 261-264) in Method section, and to explain the choise of this method instead of other, already used in Content Analysis. For example, was it already used in similar investigation? 

See for example the following description of the method (I suggest to see how Authors used findings to esxplain Results): 

Xiong, Y., Cho, M., & Boatwright, B. (2019). Hashtag activism and message frames among social movement organizations: Semantic network analysis and thematic analysis of Twitter during the# MeToo movement. Public relations review45(1), 10-23.

This study employed multiple methodological approaches, including semantic network analysis, thematic analysis, and correlation test. First, semantic network analysis was used to answer RQ1 that examines words and the linguistic structure co-created by SMOs and publics in the #MeToo movement. More specifically, semantic network analysis was used to analyze the ego-network structures of “feminism.” As a type of social network analysis which examines the textual data as the unit of analysis, semantic network analysis focuses on the “overall semantic content of the social meaning” (Golob, Turkel, Kronegger, & Uzunoglu, 2018, p. 4). Semantic network analysis allows researchers to understand the most frequently mentioned symbols (Doerfel & Barnett, 1999), to identify the dynamics of conversations in social networks (Yang & Veil, 2017; Zavattaro, French, & Mohanty, 2015), and to reveal the structure of texts by measuring co-occurrences of words (Danowski, 1993; Doerfel, 1998). Semantic network analysis is an appropriate approach to understanding how meanings were co-created by SMOs in terms of feminism as it analyzes “the structure of a system based on shared meaning” (Doerfel & Barnett, 1999, p. 589). More specifically, ego-network analysis provides a network between ego (feminism-centered words: feminism, feminist, and femicide in this study) and a set of words that are directly related to the ego (Everett & Borgatti, 2005). One limitation of semantic network analysis is left out some rich and in-depth details of the texts (Yang & Veil, 2017). Schultz, Kleinnijenhuis, Oegema, Utz, and van Atteveldt, (2012) asserted that providing rich qualitative descriptions of the underlying meanings of the words is necessary to reduce the limitation of semantic network analysis. Thus, the researchers conducted further close examination of the texts through thematic analysis to better understand themes emerged from Tweets co-created by SMOs and publics in the #MeToo movement. A thematic analysis was employed to address RQ2 that explores the frames included in SMOs’ hashtags. Thematic analysis focuses on latent themes and assumes that meaning and experience are socially produced and reproduced (Braun & Clarke, 2006). Last, a correlation test was used to explore the relationship between the number of hashtags and frequency of retweets.

--------

Liu, W., Lai, C. H., & Xu, W. W. (2018). Tweeting about emergency: A semantic network analysis of government organizations’ social media messaging during hurricane Harvey. Public relations review44(5), 807-819.

Using a semantic network approach to examine crisis response strategies has two significant advantages. First, methodologically, semantic network analysis supplements existing SCCT research by extending the examination of response strategies from thematic categories to associative patterns among key issues, actions, and actors. Previous research of SCCT has much relied on qualitative methods such as discourse analysis (Benoit, 1997) and manual content analysis (e.g., Kim & Liu, 2012). For example, Kim and Liu (2012) content analyzed the response messages from 13 corporate and government organizations during the 2009 flu pandemic. They identified different crisis response patterns between corporations and governments by comparing how frequently each type of organizations employed crisis response strategies, including “denial,” “diminish” and “reinforce” (p. 69). While traditional content analysis enables the comparison of response strategies across situations, it is still limited in that the coding scheme usually does not offer a close-up look at the semantic features of the messages, nor the association pattern among any emerging concepts.

Second, the semantic network approach enables more nuances to be identified when comparing the use of same response strategies across multiple crisis stages. For example, the same strategy of instructing information may focus on different aspects of a crisis or emphasize involvement of different actors. Such variations are likely driven by distinctive communication goals specific to each crisis stage. In the context of using social media for crisis management, Houston et al. (2015) identified different social media use goals across various stages of a natural disaster. In the pre-disaster stage, the communication goal deals primarily with delivering disaster preparedness and warning information, where government organizations such as city police and fire departments use social media to broadcast impending situations. At this stage, instructing information is likely to be the predominant type of strategy employed. During the disaster, the communication goals may shift from information delivery to more instrumental resource mobilization, such as requesting assistance, calling for volunteers and donations, and reporting real-time disaster response updates. At this stage, the strategy of instructing information is still widely present, but its emphasis shifts from informing to mobilizing. Therefore, it is important to distinguish different semantic-level meanings emerging from the same response strategy, as they are likely to vary as the crisis evolves.

Given the different communication goals as outlined above, we posit that the three most prominent response strategies for natural disasters—instructing information, adjusting information, and bolstering—are likely to be employed at varying degrees, and the specific issues, actions, and actors emphasized in each strategy may also evolve across stages. In the following, we detail semantic network analysis and ways of operationalizing response strategies.

-----

Doerfel, M. L., & Barnett, G. A. (1999). A semantic network analysis of the International Communication Association. Human communication research25(4), 589-603.

Semantic network analysis differs from traditional network
methods because it focuses on the structure of a system based on shared
meaning rather than on links among communication partners. In other
words, two nodes are connected in a semantic network to the extent that
their uses of concepts overlap. The meaning-centered network approach
stems from Monge and Eisenberg's (1987) call to enhance traditional network analysis by focusing on communication content. Semantic network analysis also has a theoretical foundation based on cognitive processes.
Learning theorists argue that words are hierarchically clustered in memory (Collins & Quillian, 1972). Thus, spatial models that illustrate the relationships among words are representative of meaning (see Barnett &
Woelfel, 1988). As a result, studies have turned to analysis of text with network analysis techniques (Danowski, 1982; Jang & Barnett, 1995; Rice &
Danowski, 1993; Stohl, 1993). 

Reviewer 4 Report

Thank you for clarifying and amending the manuscript.